# 3D Land Administration: A Review and a Future Vision in the Context of the Spatial Development Lifecycle

**Eftychia Kalogianni [1]** , **Peter van Oosterom [1],\*** , **Efi Dimopoulou [2]** and **Christiaan Lemmen [3,4]**

1   GIS Technology Section, Delft University of Technology, P.O. Box 5030, 2600 GA Delft, The Netherlands;
    E.Kalogianni@tudelft.nl
2   School of Rural and Surveying Engineering, National Technical University of Athens, PC 15780 Athens,
    Greece; efi@survey.ntua.gr
3   Faculty ITC, University of Twente, P.O. Box 217, 7500 AE Enschede, The Netherlands;
    Chrit.Lemmen@kadaster.nl
4   Netherlands Cadastre, Land Registry and Mapping Agency, Kadaster, P.O. Box 9046, 7300 GH Apeldoorn,
    The Netherlands
\*   Correspondence: P.J.M.vanOosterom@tudelft.nl

**Abstract:** Land Administration practices worldwide rely mainly on 2D-based systems to define legal and other spatial boundaries related to land interests. However, the built environment is increasingly becoming spatially complex. Land administrators are challenged by an unprecedented demand to utilise space above and below earth's surface. The relationships between people and land in vertical space can no longer be unambiguously represented in 2D. In addition, the current societal demand for sustainability in a collaborative environment and a lifecycle-thinking, is driving the need to integrate independent systems with standalone databases and methodologies, associated with different aspects of the Spatial Development lifeCycle (SDC). Land Administration Systems (LASs) are an important component of the SDC. Today, a LAS is often mandated and managed as a domain in isolation. Interaction and data reuse with the other phases of the SDC is limited and far from optimal. It is expected that effective 3D data collaboration, sharing, and reuse across the sectors and disciplines in the lifecycle will enable new ways of data harmonisation and use in this complex environment; will improve efficiency of design and data acquisition, as well as data quality (in relation to specific regulations); and will minimise inconsistencies and data loss within information flows. Overall, a cross-sectoral approach is directed towards improving the current state of the Land Administration (LA) domain. This paper consists of two parts. In the first, a review of the current situation, with respect to LASs is presented, concluding the needs for improvement in terms of effectiveness and consistency. In the second part, the vision for the future of LASs is introduced in a wider context, and as an important phase in the SDC, with regards to legal, technical, and organisational aspects. In this part, the needs and considerations that result from the evolving environment and the emerging technological advances are addressed, with a view to discussing a cross-sector approach to collect, maintain, reuse, and share 3D data. In such a cross-sectoral approach, various interoperability issues appear, making it necessary to introduce and use standards. In this respect, the ISO 19152:2012 Land Administration Domain Model (LADM) in its current Edition I, as well as in Edition II (expected in 2022) may serve as the standardised core structure of a 3D LAS, with respect to its role as further presented in this paper. In parallel, the evolution of the Building Information Modelling (BIM) in the design and construction industry, as well as the fact that BIM plays a central role in the life cycle of development projects, are well recognized. Emphasis is given on feasible reuse of BIM/IFC (Industry Foundation Class) data in a 3D LAS. Those considerations are addressed through a web-based system architecture for a future 3D LAS, thereby attempting to integrate heterogeneous systems in the SDC.

**Keywords:** 3D; Land Administration System (LAS); Land Administration Domain Model (LADM); Building Information Model (BIM); spatial development lifecycle; standardization; interoperability

## 1. Introduction

Section 1.1 provides background material and introduces key concepts facilitating the understanding of this manuscript and its views in relation to Land Administration (LA), while Section 1.2 provides the methodology followed.

### 1.1. Land Administration

Over the last 15 years, a number of political, economic, environmental and social factors as well as technological innovations have profoundly changed the outlook for efficient management of land, water, natural resources, and the built environment. Security of tenure and registration of property rights ('property rights' should be taken in the broadest context – in principle all relationships between people and land are covered by this term. Implementations can have variations) are recognized as important components for achieving sustainable development in a global context (in view of the Sustainable Development Agenda 2030 [1]), particularly in urban areas [2].

Land Administration informs the 'how', the 'what', the 'who', the 'when', and the 'where' of land tenure, land use, land value, and land development [3]. It is an inter-disciplinary field, involving experts and knowledge regarding legal and technical aspects, with institutional support to establish relationships between involved parties, and with documented requirements for data acquisition methods, modelling approaches, data management, and visualization methods. Land Administration is described as the "process of determining, recording and disseminating information about the relationship between people and land" [4]. In this context, the role and functional requirements of Land Administration Systems (LASs) have significantly evolved over the years, while land tenures are increasingly being created with explicit limits in the third dimension [5].

In this paper the term '3D Land Administration' replaces the term '3D Cadastres' as used by the International Federation of Surveyors (FIG), over a series of Workshops organized by the "Joint FIG Commission 3 and 7 Working Group on 3D Cadastres", starting in 2001, all under the name '3D Cadastres' with key overviews published [2,6]. The motivation is based on the definition of Land Administration used by the International Standards Organization (ISO), which includes the 3D representations [4] in the standard ISO 19152:2012, Geographic Information–Land Administration Domain Model (LADM). The definition used in ISO 19152 Edition I (and will be extended in Edition 2), is re-formulated from the definition of land administration as stated in the land administration guidelines as from UNECE 1998 [7]. The term 'Land Administration' is used in these guidelines to refer to the processes of recording and disseminating information about the ownership, value and use of land and its associated resources. This concerns Land Registry and Cadastres. The definition of Land Administration in LADM is derived from this definition.

A second reason why in this paper the term 'Land Administration' is used is that it is less ambiguous than the term 'Cadastres', which in some parts of the world implies a focus on the spatial aspects. However, with the term 'Land Administration' both the legal (administrative) and the spatial aspects are covered – indicated as Land Registry and Cadastres. In this paper Land Administration concerns Land Registry (including restrictions as a result from spatial planning) and Legal and Tax Cadastre.

LASs support the functioning of land markets in an efficient way and are, at the same time, concerned with the administration of land as a natural resource to ensure its sustainable development [8]. Further, it is worth noting that LASs contribute to facilitating digital economies, fundamental datasets, and smart sustainable cities of the future [9]. However, as already mentioned, the majority of existing LASs around the world are currently based on 2D systems where a 2D parcel (spatial unit in LADM

terminology) is the key-entity of property registration. Those systems are, by nature, supported by processes that are designed for 2D parcel representation in digital format and are often still implemented using paper-based records. Nevertheless, in so far as they delineate the extent of land, water, air and underground interests they are inherently 3D. In order to cope with the societal trends, such as urbanization, societal disparities, and the digital transformation, those systems need re-engineering to extend into 3D, as stated by a large number of publications in the field ([2,10–15].

One of the key drivers to move forward towards 3D registrations in LAS is the need from the real world to align to technological developments. Presently, technologies to collect, store, maintain, visualize and disseminate 3D information are mature and becoming mainstream. This refers to advanced 3D data surveying/acquisition techniques, availability of detailed Building Information Models (BIM), 3D web visualisation platforms, 'Smart Cities' applications, etc. Public use and expectation of 3D information is high. It is higher even than the legal mandate in several countries, which makes it relevant to look into the future of 3D LAS in a wider context.

Summarising this overview, it is apparent that 3D LAS, in its broader concept, is a quite inter-disciplinary field involving experts and knowledge regarding legal aspects (e.g., how to define and register a 3D parcel), institutional support to establish relationships between involved parties, and technical support to realise it (data acquisition methods, modelling, storage and visualisation techniques). In this respect, organisations responsible for Land Administration around the world recognise the need to advance the practice of property registration by adopting current technological trends, and are taking steps forward to register multi-level property rights in such a way that the registration provides a clearer insight into the (3D) legal situation [16]. However. the level of sophistication of each 3D LAS in a jurisdiction will in the end be based on the user needs, land market requirements, legal framework related to each jurisdiction, strategic and planning policies, as well as technological options.

## 1.2. Reusing BIM and GIS Models for 3D LASs

Much of the current research in the field of Spatial Information Science focuses on issues related to 3D geoinformation: techniques for data collection, data management, optimizing processes, web-based data dissemination and visualization, standardization of 3D information, and interoperability of solutions. Particularly, 3D modelling is expanding its application in the built environment. This ranges from the design of individual buildings using digital engineering tools such as BIM to the city level (Smart Cities). In the latter, CityGML applications and 3D Geographic Information Systems (GISs) comprising photorealistic 3D models of natural, rural and built environment (including structures above and under the ground) are the most dominant solutions. However, the borders between those applications are breaking down as the world is increasingly migrating towards data integration. There is a need to combine independent databases in systems associated with different disciplines, aspects and scales of the (built and natural) environment.

Moreover, research is being carried out in the field of linking LA information to 3D digital representations (usually of the urban environment). Specifically, the reuse of 3D digital models such as BIM and 3D GIS to define and visualise the spatial properties of 3D LAS is currently being investigated [17–22]. Such source data can be expected to have capabilities to specify semantics, which can identify property units accurately, represent cadastral boundaries better, and visualise complex buildings in more detail [17]. BIM is an important and promising development in the Architecture, Engineering, and Construction (AEC) industry for both the modelling approach (BIM) and the output product (BIM – 3D AEC models). In this paper the term 'BIM' is used for the products, the 3D AEC models, and its evolution towards integrated sustainable design [23]. Dissemination of information is highlighted. BIM has revolutionised the design and construction industry around the world in recent years. It is being adopted rapidly as more BIM data are generated and becoming available.

In this context, Liu et al. [24] underlined that although BIM can provide much detailed information for LAS purposes, sometimes this information can be too detailed, and a simplification process is

required, while information concerning ownership and transaction history, is not available in BIM. Moreover, BIM focuses on the building element properties of a single or complex building, while city models based on CityGML (or similar standards), focus on buildings' composition within the urban fabric, which may also be suitable for 3D LAS applications. Recognising that BIM and GIS models can be used as complimentary input data for LAS, recent research [17] proposes to use both BIM and CityGML for LA purposes based on LADM. There is much interest at an international level in the reuse of information from BIM and GIS environments as source data for LA purposes and other applications, also embodying the concept of the lifecycle of information.

### 1.3. Methodological Approach

This first objective of this paper is to provide a brief description of the current state of LASs worldwide, discussing the present situation between the various phases and disciplines involved in the SDC and referring to the standards that are in use today. LASs are viewed in a broader context in terms of relations with the various types of 3D objects. The second objective of the paper is to introduce and discuss a vision for 3D LAS in the future, based on current trends, requirements, and considerations that arise from the constantly changing environment. This work highlights and addresses the need to move from 2D-based LAS systems to 3D LAS, within the lifecycle thinking, and highlights the potential for reusing Industry Foundation Class (IFC) data as source information for a 3D LAS. BIM/IFC enrichment with legal information does not only affect the geometric and modelling complexity of input data and its quality, but also the reusability of information within the spatial development lifecycle.

At the previous section, the technological advances that may support such a 3D LAS have been briefly described (and are further discussed in Section 2.3). It is highlighted that IFC, as a semantically rich formalism and the most common publication format for BIM, is considered a promising source for semantically enriched spatial data regarding LA in an urban environment, including buildings, apartment rights and infrastructure elements.

3D LAS could benefit from the lifecycle thinking, by reusing geometries from earlier phases of the SDC (specifically: design and obtaining permits). Therefore, the focus of this paper is also on BIM/IFC, while the other formats (such as CityGML) are not used, or are much less used in these early phases.

IFC has been chosen to be linked and to provide input for a 3D LAS for the following reasons:

■ It is a recognised as ISO standard [25,26];
■ Its lifecycle is more and more used in AEC and design stage;
■ Recently BIM has also started to evolve at the permit process (e.g., a new strategy for BIM has been announced in Dubai [27] that enables a faster and more efficient building permit system);
■ It occurs earlier in the spatial development lifecycle rather than other standards (e.g., CityGML);
■ There is a constantly increasing number of BIM models, etc. becoming available;
■ It is considered as a strategic enabler for improving decision-making and delivery for both buildings and public infrastructure assets across their whole lifecycle [28].

The work presented in the second part of this paper, mainly builds on and evolves concepts from previous research projects, such as Cemelini [29] and Meulmeester [22]. Moreover, the literature review incorporates the results of the analysis of the most recent "3D Cadastres Questionnaire" [16] referring to the challenges and expectations for the future of (3D) LAS in the various countries involved.

In order to support and validate this visionary 3D LAS, a system architecture of a future LAS based on the principles of data reuse and interoperability is proposed. It is a web-based system consisting of four-components: data acquisition from various sources, data processing and validation, data storage and management, and data visualisation and dissemination, presented in detail in Section 4. A web-based system architecture is selected to connect the heterogeneous systems involved in this lifecycle flow. It enables maximum dissemination at the last stage. This system will be used to validate the approach in terms of applicability and data loss.

To continue, the paper is structured as follows: its "review" character continues with Section 2, which presents the concept and characteristics of the spatial development lifecycle, as well as with part of Section 3, which discusses the current situation of 3D LAS worldwide and presents the requirements for a future 3D LAS. Therefore, Section 4 briefly presents the vision of a system architecture and prototype of a web-based LAS reflecting the vision for the future of LAS, while the last sections are dedicated to Conclusions and Future Work.

## 2. 3D Spatial Development Lifecycle

This section introduces the concept of spatial development lifecycle (Section 2.1), being driven by the current societal demand to improve sustainability performance through collaboration, the need to integrate independent systems associated with different aspects and at various scales of the spatial development lifecycle, and the phases of the processes that exist today. Section 2.2 underpins the need to combine independent systems, methodologies, and procedures of this lifecycle and highlights current incompatibilities and interoperability problems. The approach to tackle these interoperability issues is standardisation, as presented in the last Section 2.3.

### 2.1. Phases of the Spatial Development Lifecycle

The built environment encompasses associated interdisciplinary aspects of design, construction, management, and operation of the created surroundings and artefacts. The key industry sectors directly concerned with those aspects include AEC, as well as Geography, Land Administration, and Urban Planning. Although interwoven in certain aspects, these disciplines rely on different systems in the synthesis and management of information associated with the built environment. In practice, thos disciplines are mutually affected. Progress in the integrated use of the data sets has proven to be slow and expensive, with inconsistencies and duplication in representation of the same objects through different phases of their lifecycle, resulting in mistakes and ambiguities.

This does not only apply to the objects of the built environment that already exist, but also to those that are in the design process. For instance, when construction of a new building is planned, it will follow the spatial development lifecycle stages: zoning according to relevant regulations and constraints, field surveying, designing, permitting, financing (if relevant), constructing, registering in the land administration database, maintaining, and demolishing.

Likewise, this also applies to other objects that are not encountered as elements of the built environment, such as agricultural areas and natural resources, including *inter alia*: forests and forestlands, marine spaces, shores, air parcels, minerals, mining areas, and other under and above ground utilities. Eventually, different stakeholders should share and exchange information during the whole life cycle in order to represent complex boundaries. Today, data in the built environment is rarely shared between actors and between the phases of SDC, due to technical, legal, cultural, and business reasons.

Collaboration across different stakeholders in the Land Administration domain is expected to enable new ways of data harmonisation and use in this complex environment, to improve efficiency of design and data acquisition, to improve data quality (in relation to specific regulations), to minimise inconsistencies and data loss, mismatch and overlap between the various stages, and to enhance data re-use from design phase to end user and registration/operation phases. A cross-sector approach to collect, maintain, reuse, and share 3D data can improve the efficiency of current situation, while data become suitable for various new and existing applications.

Specifically, the disciplines that are presently involved in the different phases of the spatial development lifecycle (Figure 1) operate quite autonomously, using custom-made, independent methodologies, software and workflows. It should be considered that financial data, permit data, occupancy status, maintenance history, and other information are fundamental aspects in the spatial development lifecycle and should be maintained and effectively exchanged during its various phases. The role of Land Administration in the Spatial Development Lifecycle is particularly linked with the

process of registration, however it also plays a (larger or smaller) role in each one of the other phases. Resolving issues on data sharing and data integration will increase effectiveness in the spatial lifecycle development by the provision of an efficient, well-organised data flow based on standards. This is essential, especially for wide large-scale reuse in complex environments.

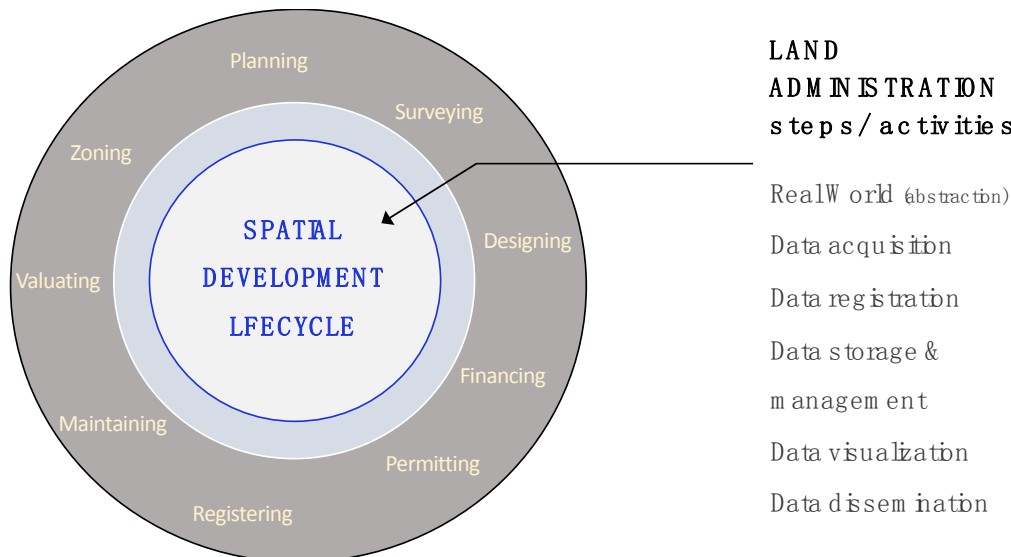

**Figure 1.** Spatial development lifecycle.

## 2.2. Need for a Structured Data Flow for Cross-Sectoral Collaboration

Shaping and sharing AEC, spatial and economic data into an efficient data flow represent a challenge. The potential for the reuse of information within the spatial development lifecycle is a significant factor in calculating its economic value. By avoiding inconsistencies/mistakes and by adding real world coordinates, the value and types of data are increased for all stakeholders. When information is shared between the phases, additional information such as lifecycle information, versioning information, and unique identifiers, are needed to achieve a more process-oriented approach to the information flow [30].

Various disciplines working in the spatial development lifecycle have their own view and interpretation of its importance, use, and application; they have unique vocabularies and are quite autonomous, using custom-made procedures. The divergent phases and stakeholders during the lifecycle of an object, highlight the issue of efficiently connecting the different domains and ultimately delivering the right piece of information to the right party at the right time: leading to effectiveness.

It is noted that a crucial phase of this lifecycle is the registration of the object in a cadastral database, and hence it is vital to consider workflows to exchange and reuse this information during the various phases. Towards a holistic lifecycle approach, the design of a structured data flow for cross-sectoral collaboration is of crucial importance.

One of the significant concerns in this direction is the data and specifically its quality, source and dimensionality. Given that (spatial) information comes from many different sources and is managed by a (large) number of different providers, there is an overwhelming requirement to easily discover and share this information. Spatial data may originate from a recent survey e.g., using laser scanner (point cloud), using Unmanned Aerial Vehicles (UAV), or using Global Navigation Satellite System (GNSS) receivers (using GPS Galileo navigation data), while data may also be provided from other databases or use other design drawings or BIM models as source.

When exchanging and sharing such data within the various processes that occur in the SDC, it is important to set criteria to evaluate the quality of the data, so that it is suitable to the purpose of each application/phase. Data quality aspects are to be considered for the data collected using

various acquisition techniques, for the data reused from the design phase, as well as for the non-spatial data reused from existing databases (land administration, land use, valuation, etc.). According to international and national standards, spatial data can be evaluated as to whether it is acceptable or not using geodata quality parameters such as completeness, logical consistency, position uncertainty, thematic uncertainty, temporal uncertainty and usability [31]. It is important to notice that the quality of input data, reflects on the whole SDC.

The data is often 2D, lacking 3rd dimension/height information entirely, or 2.5D, i.e. featuring height as an attribute to horizontal position/plane rather than as an independent coordinate. Furthermore, the vertical dimension may be sparse with height measured at few locations only, and it may be ambiguous because it is not always clear whether the values represent height relative to a specific surface with unknown elevation or height relative to an established height datum. It is often also unknown whether the data represents the current situation, the possibly different as-built state, or just the as-designed state. Furthermore, its geometric accuracy and completeness is often unidentified. Much of the attribute information, as well as its history/versioning information required to support specific applications is not available or not represented at the appropriate level of detail.

Data sharing means the data is collected once and used many times through establishing linkages (for example through Spatial Data Infrastructure (SDI)) [32], as well as collected for one purpose and subsequently used for another. Thus, duplicated efforts in data collection and maintenance can be avoided. For instance, spatial data regarding a road alignment may be collected and/or surveyed in order to produce a road map. This spatial dataset can then be used by someone else to estimate city zoning regulations. External links to other databases (e.g., addresses, population register, business register, building register, utilities register, etc.) are needed in all sectors to source input data and/or disseminate results and to address interoperability issues via standardised approaches and exchange formats. Multiple encoding and exchange formats are used to store this information. Standards have a key role in this respect and are essential to delivering authoritative geo-information services and products which meet the requirements of the wider community of users [33]. All the involved stakeholders in the different lifecycle phases will benefit from 3D datasets, either when representing a real-word model or a design of planned/future scenario, e.g., architectural plans, spatial plans, etc. Simultaneously, 3D datasets are becoming ubiquitous for decision-making and for improving the effectiveness of governance at different levels. Involved parties will become data producers themselves (for a mix of 2D, 2.5 and 3D datasets) and there is need to adopt bottom-up and top-down governance approaches, regarding data acquisition and storage, data processing and sharing from different heterogeneous sources, by working with standards. Much effort is made in the AEC and GIS domains to address interoperability issues via standardised approaches and exchange formats, as presented in the following section.

In the same direction, to support a product through its life, the ISO standard 10303-239 Product Life Cycle Support (PLCS) addresses the key challenge of how to keep information needed to operate and maintain a product aligned with the product throughout the inevitable changes that occur in the course of its life cycle [34].

### 2.3. Importance of Standards

A 3D LAS covers both built environment and non-built environment elements, e.g., subsurface natural resources, airspaces, etc. Nonetheless, the urban environment must address multiple scales of spatial information [35] originally developed for different purposes. From geographic information, to civil engineering information, to BIM as basis for accurate and comprehensive spatial modelling for Smart Cities and SDI, even for Spatial Information Infrastructure (SII).

Several organisations, industry consortia and communities are involved in standards' development activities related to (3D) geoinformation; to name a few: ISO TC/211, Open Geospatial Consortium (OGC), European Committee for Standardisation (CEN), World Wide Web Consortium (W3C), Web 3D Consortium (W3D), BuildingSMART Alliance, 3D Industry Forum (3DIF), Open Design Alliance,

Khronos group, etc. ISO TC 211 and OGC are considered the two dominant ones in the geoinformation field, employing processes and approaches which aim to ensure the development of international standards with a wide scope. Their aim is to ensure the ability to integrate datasets and related services of different types and from different sources, minimising costs and problems, while reducing dependence on implementation specifics (software, etc.).

Currently, a wide range of standards related to 3D is available and in principle each one has been developed for a specific purpose. Such standards are related to data models, data exchange and storage formats, data dissemination through encoding formats, and/or web services. An extensive comparison of such standards has been performed by Zlatanova et al. [36], based on selected criteria. The most prominent open standards in the geoinformation domain are: the OGC standard CityGML [37] for storage and exchange of 3D city models, the international IFC standard [25] for BIM models, the OGC standard LandInfra [38] and its GML-based encoding, InfraGML modelling and representing land and infrastructure features. There are several studies that investigate the interoperability between those standards, as well as a recent research by Kumar et al. [39] that analyses the differences and similarities between those three standards, with regards to certain criteria as geometry, topology, semantics, encodings, etc. With respect to the legal and administrative information, the most dominant standards are the ISO standard LADM [4] and e-Plan, mostly used in Australia, New Zealand and Singapore. At the following paragraphs, a brief description of IFC and LADM is presented, as those two standards are used at the future 3D LAS, as presented in Section 4.

### 2.4. Building Information Model

One of the most dominant standards in the AEC is the BIM, which is defined by international standards as "shared digital representation of physical and functional characteristics of any built object [ . . . ] which forms a reliable basis for decisions" [26]. BIM refers to virtual 3D building models containing 3D digital spatial information as well as semantic information about a building to support decision making throughout its lifecycle [40].

BIM is being adopted rapidly in different parts of the value chain as a strategic tool to deliver cost savings, productivity and operations efficiencies, improved infrastructure quality and better environmental performance [28]. Recognising that the moment has now come, the EU BIM Task Group has been established in order to build a common European approach for accepting and adopting BIM. In this direction, national and governmental BIM councils are initiating policies and strategies in various countries (Ireland, Germany, UK, United Arab Emirates, The Netherlands etc.), resulting in increasing development of detailed BIM models. In some of those countries, the use of BIM is already under a government mandate (UK, The Netherlands, etc.) for certain projects (i.e. in Germany for transportation projects). Various countries, by including BIM requirements in public procurement, play a key-role in accelerating significantly the early stages of BIM acceptance, adaptation and implementation.

BIM is not new, but it is a global trend that is growing. The term 'BIM' was first mentioned in 1992 by van Nederveen and Tolman [41], as a way to model multiple views of buildings through decomposition. Since then a lot of progress has been made. Nowadays, as stated by the EU BIM Task Group [28], the social, environmental and economic benefits of digitalization are well recognized: BIM is a digital form of construction and asset management. It is a strategic enabler for improving decision making to manage buildings and public infrastructure assets across their lifecycle, bringing together technology, process improvements, automation, and digital information. Figure 2 illustrates the application of BIM along the construction value chain.

BIM models are rich in geometry, semantics and topological information. The BuildingSMART alliance has developed various international open standards for storage and exchange of different aspects of the building information, namely: IFC, IDM/MVD (Information Delivery Manual/Model View Definitions), BCF (BIM Collaboration Format) and IFD (International Framework for Dictionaries) [42].

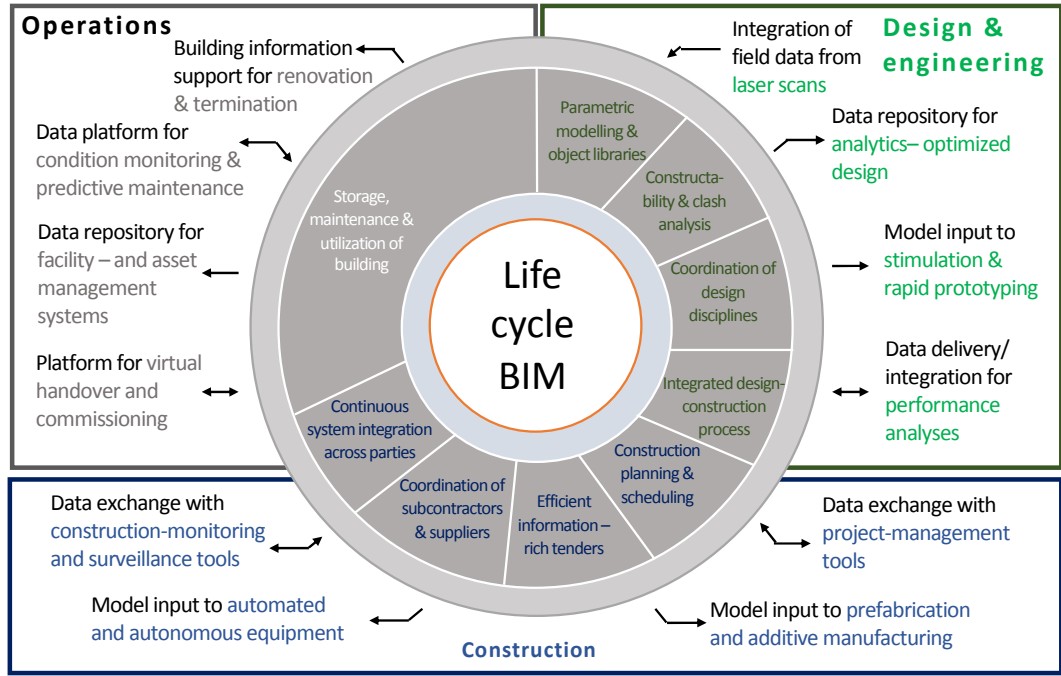

**Figure 2.** Application of BIM along the construction value chain ([43], adapted).

IFC is an industry-specific data model schema, the most common building information exchange format and international standard [25]. All physical building elements can be modelled, stored, and managed hierarchically in the IFC standard, which makes it easy to exchange building information for multiple purposes in different BIM platforms [17]. Data can be exchanged in platform neutral, open file formats that are not controlled by a single vendor or group of vendors. One commonly used collaboration format for BIM is IFC. The IFC model specification is open and available, it is registered by ISO and is an official International Standard ISO 16739-1:2018 (the previous version was ISO 16739:2013). IFC files can contain many types of classes. The geometry of BIM models in the IFC format can be represented using Constructive Solid Geometry (CSG), Sweep Volume, or Boundary Representation (B-Rep) [42]. Elements are modelled in local coordinate systems defined by a hierarchical set of transformations that correspond to the levels in a decomposition structure (typically a site, project, building and individual floors).

The IFC defines data requirements for buildings over their life cycle, represented as an EXPRESS schema and an XML schema (XSD) [25]. It can be encoded in various encoding formats, such as STEP Physical File (SPF), XML and JSON. BIM aims to play a central role in the life cycle of developments. As BIM/IFC are rich geometry models, they can be used in compliment with LADM, which contains -among others- legal information (as described at the next section). BIM is being used more and more and it is considered a promising source of semantically enriched information. Research in the academic community investigates various methods for using BIM/IFC as a source in the Land Administration domain, especially to apartment complexes. But, also, to other object's types; e.g., tunnel or underground parking. Currently, BIM are widely used as modelling sources, providing detailed physical and semantic information for buildings that can be further used to identify and represent 3D property boundaries accurately.

BIM/IFC data is considered and important source of information for the proposed system architecture of a future 3D LAS, as presented in Section 4.

## 2.5. Land Administration Domain Model (LADM)

The LADM is a conceptual model and one of the first spatial domain standards within ISO TC211, aiming to support "an extensible basis for efficient and effective Land Administration System

development based on a Model Driven Architecture (MDA)" and to "enable involved parties, both within one country and between different countries, to communicate based on the shared ontology implied by the model" [4]. LADM is based on user needs and provides standardised terminology enhancing interoperability between information systems. The standard is capable of supporting the progressive improvement of Land Administration and can potentially be used to support organisational integration [44], for example, between (often distributed) land registry and cadastral agencies.

The growing recognition and influence of the standard is revealed by the multiple country profiles that have been developed; several LADM implementations through technical models and encodings; as well as parallel activities, such as development of Land Administration domain ontology, support of strata titles, etc. Additionally, with the increasing need for 3D land administration information, LADM has been used widely around the world as it supports the 3D representations of spatial units without adding any additional burden to the existing 2D representations [4].

The revision of the standard started in 2019 and it will be a joint activity, supported by many organisations and institutions. The ambition is to go beyond just a conceptual model by providing steps towards implementations (e.g., more specific profiles, technical model in various encodings, etc.). The intention is that future editions of LADM should be backwards compatible with earlier editions. Figure 3 illustrates the progress of the LADM standardisation project.

The second edition of the standard, taking into account the spatial development lifecycle concept aims to [45]:

■　extend the initial scope of the conceptual model to include the following concepts: valuation information, spatial planning/zoning, land administration indicators related to the Sustainable Development Goals, linkage of legal objects with physical ones, indoor models, support of marine spaces, and support of other legal spaces: mining, archaeology, utilities, etc.

■　improve the current conceptual model, including: formal semantics/ontology for the LADM Code Lists, more explicit 3D+time profiles, an extended survey and legal models, etc.

■　include technical implementation through the most dominant encoding standards: BIM/IFC, CityGML, LandXML, LandInfra, IndoorGML, GeoJSON, etc.

■　include process models for survey procedures, map updating, and transactions (e.g., blockchain).

The second Edition of LADM will be organized into multiple parts. Alternative Working Titles of the packages (or parts) are as follows [45]:

■　Part 1 – Land Administration Fundamentals

■　Part 2 – Land Tenure or Land Registration or Land Interests

■　Part 3 – Marine Space or Marine Geo-Regulation

■　Part 4 – Land Valuation

■　Part 5 – Spatial Planning

■　Part 6 – Implementations (including Link with BIM and other technical encodings (RDF, CityGML, InfraGML, INTERLIS, GeoJSON, processes, etc.).

## 3. 3D Land Administration Systems: Current State and Future Vision

Since the inception of research on 3D LASs worldwide, about 30 years ago, the world has changed significantly, and this also reflects on the progress and advancements of 2D and 3D LASs. Looking back, the systems in use were often manually maintained, paper based and completely dedicated to the registration of land and RRRs (Rights, Restrictions and Responsibilities) [46].

As stated by Steudler [47] the term 'land' should be interpreted in the broad sense, also including water bodies (rivers, lakes, seas, oceans) and spaces above and below the surface, that is, air space and subsurface spaces. Land administration comprises an extensive range of systems and processes to administer: Land Tenure, Land Value, Land Use and Land Development, which are interrelated and influence each other. This global approach to LASs is presented in Figure 3.

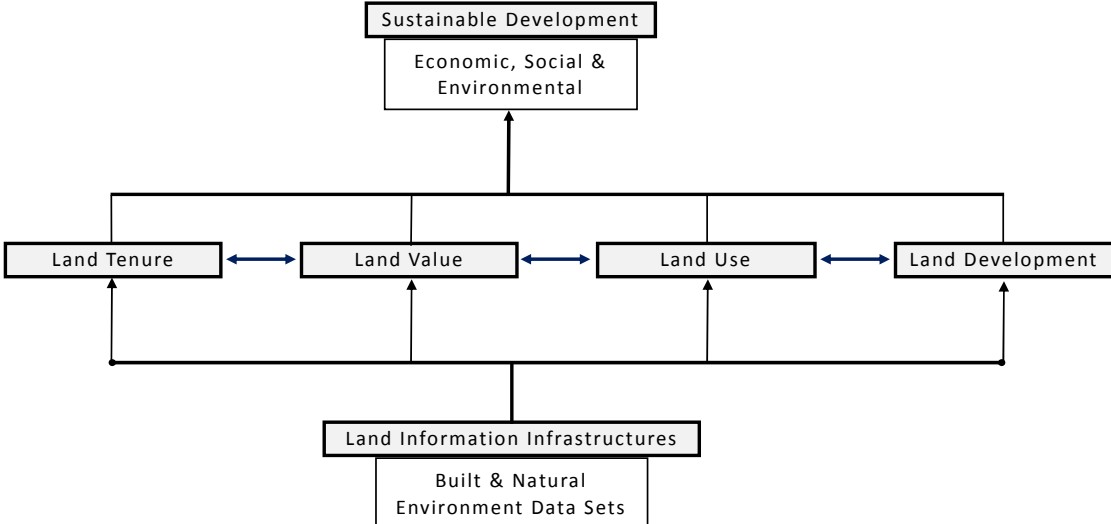

**Figure 3.** A Global Land Administration Perspective ([48], adapted].

This section briefly presents the current situation and latest approaches of 3D LAS worldwide (Section 3.1), and the various types of 3D spatial units that are physically identified and legally recognised in various jurisdictions worldwide (Section 3.2). Moreover, it introduces the vision of future 3D LASs, by identifying the requirements and challenges for a well-operational 3D LAS (Section 3.3).

### 3.1. Current State of 3D LAS Developments Worldwide

Until today the 2D parcel has been the main entity in property registration in most jurisdictions worldwide, however infrastructure density leads to complex interleaving triggering legal, organisational, and technical challenges [11,14]. The increasing complexity of infrastructures requires proper registration of properties' legal status, and thus the 2D cadastral systems are increasingly facing challenges in recording, managing, and visualising the spatial extent of cadastral spaces [10–12,14,49].

In the last decade, the number of partial implementations of 3D parcel registrations around the world has increased significantly [2,14,15], taking advantage of the developments supporting the third dimension in the field of GIS technology. A significant number of studies have been carried out to establish 3D LAS solutions to improve the registration of multi-level properties.

Specifically, several jurisdictions, including the Netherlands, Sweden, Czech Republic, Croatia, Singapore, the city of Shenzhen in China and the Australian states of Victoria and Queensland, have examined and implemented prototype 3D LAS as systems for the comprehensive documentation of land and property information [50]. The implementation of a well-functioning 3D LAS is still a challenge in all those countries, as there are legal, institutional, and/or technical shortcomings and challenges that need to be addressed.

So far, no country has a complete, operational 3D LAS incorporating all those aspects, however there are several jurisdictions which do have operational and efficient solutions supporting at least partly the context of 3D LASs as described above. Those developments can be mainly categorised as "fully operational" implementations applying a holistic approach achieved at different levels of maturity and "partly-operational" implementations exploring the process of developing a 3D LAS focusing on different aspects; e.g., submission of 3D survey plans, prototypes linking legal spaces with physical models, implementations that focus on 3D visualisation, and implementations that focus on (3D) constraints and validation rules [51]. In several states of Australia, the legislation supports either strata titles and/or volumetric parcels and for over 30 years survey plans have been submitted with these 3D descriptions [52]. However, the database with parcels is still 2D. The first 3D cadastral registration of multi-level ownerships rights has been accomplished in The Netherlands, in 2016 [53], as a result of many years of research and development. This was without any change to the law and no

3D geometry in the database to describe the parcels (just a 3D drawing as part of the deed submitted in pdf). The city of Shenzhen in China [54] and Singapore [55] are close to achieving a fully operational 3D LAS.

LADM plays a key role in the advances on the 3D LASs worldwide, and several jurisdictions have adopted it since its recognition as ISO standard. Multiple LADM-based country profiles have been developed, based on the requirements of the local cadastre and/or land registration system, as well as the legislative framework. A recent work by Kalogianni et al. [56], provides a list comprising most of the developed LADM country profiles so far and reflects on those advances, with a view to providing a flexible methodological framework to build LADM based LASs. With this regard, the current situation presents a trend that more and more countries are willing to examine the feasibility of adopting LADM as a core base for their LAS. The proposed future LAS prototype presented in Section 4, builds on this statement and proposes that the core database will be LADM based and compliant.

The pace of the transition to a 3D LAS, from an existing 2D LAS, or even when there is not yet a LAS established, depends on various aspects (presented in Section 3.3). At the same time, there is a significant differentiation between the pace of integration of technological solutions in the field of 3D LAS in various jurisdictions, associated with the flexibility of legislation, differences in the conceptual apparatus, national, and technical features.

### 3.2. Types of 3D Objects and their Modelling Complexity

Cadastral parcels range from 2D to 3D collections of spaces around the world and parcel representations are defined at multiple levels of sophistication [52], depending on the available data, the regulations of each jurisdiction, the land market requirements, etc. The complexity of representations of volumetric spatial units worldwide is highly variable, like the types of 3D objects.

3D spatial units that commonly appear in the various jurisdictions around the world are the starting point for their categorisation and modelling. Research carried out in this field [2,16,51] highlights the following categories of 3D objects, which refer to underground or above ground properties, or the land/water surface (it is noted that not all of these 3D objects can be found in a well-established LAS around the world):

■　Simple 2D parcels,
■　Simple 3D parcels,
■　3D Buildings,
■　Condominiums/apartments,
■　Utility networks (oil, gas, water, electricity, telecommunications, etc.),
■　Other underground objects (e.g., parking garage, storage areas, cellars, etc.),
■　Infrastructure elements (e.g., roads, metro lines, etc.)
■　Tunnels,
■　Bridges,
■　Marine spaces,
■　Air spaces,
■　Mining spaces,
■　Natural resources,
■　Other objects (e.g., unofficial boundaries of the respective federal geo regulations)

### 3.3. Requirements and Emerging Challenges for a future 3D Land Administration System

Given this background, the requirements for a future 3D LAS are outlined in this section and the three basic directions are explored: organisational/institutional, legal, and technical aspects. Moreover, the emerging challenges that need to be taken into account for the development of a well-operational 3D LAS are presented. Those requirements derive from the analysis of the current situation and state of the LASs worldwide as presented in Section 3.1, including the modern trends in the fields of GIS

Technology, Land Administration and AEC, the UN-GGIM frameworks [57], as well as the vision for a future 3D LAS in a wider context. The formulation of those requirements has also taken into consideration the expectations for the future as stated in the latest 3D Cadastres Questionnaire [16], referring to 3D parcel representations in various formats and updated legislative frameworks.

In this respect, the need for effective Land Administration is also underlined by UN-GGIM [3], presenting nine (9) pathways for effective LAS, which are currently under development. Namely: Governance, Institutions and Accountability, Legal and Policy, Finance, Data, Innovation, Standards, Partnerships, Capacity and Education, Advocacy and Awareness. In this context, the interlinkages and integrated nature of the 2030 Agenda for Sustainable Development with its 5Ps (People, Planet, Prosperity, Peace and Partnership) [58] find direct resonance with effective land administration and management, realised through integrated geospatial information, for land tenure, land value, land use, and land development. Those pathways are based on the IGIF, the UN-GGIM Integrated Geospatial Information Framework, which provides direction in three main areas of influence: governance, technology, and people [57].

In this context the vision for a 3D LAS as a core component of the spatial development lifecycle, fits well in the Framework of effective Land Administration according to UN-GGIM [3]. Below, the vision for a future 3D LAS is annotated with respect to the 9 UN-GGIM pathways:

- the aspects of Governance, Institutions, and Accountability are involved as the vision for 3D LAS, to improve cross-collaboration between the sectors,
- it is recognised that land law and policy form the basis for LAS, and that to serve the needs of such a workflow and a future 3D LAS, they must be revised accordingly,
- the financial aspect of LA is acknowledged and an information flow proposed that will reduce the cost of current situation,
- attention is given to (spatial and non-spatial) data reuse and sharing,
- innovation can be driven by technological push and specifically the advances in the geoinformation field that can be used within the proposed approach,
- standards play a key role in this approach; namely ISO 19152:2012 LADM, which is used as the core model of LAS; while source data is expected to be in a standardised exchange format (i.e. IFC) and the dissemination approaches is also expected to follow standardised techniques.
- partnerships might variously include the creation and harnessing of strong relations within and between public sector, private sector, academia, civil society, professional bodies, coordinating organizations, and international agencies and societies [3].
- the development of enduring knowledge and skills transfer at the required level, for all stakeholders, is crucial for the smooth cooperation between the sectors, and needs to be strategically included and implemented in the context of this approach,
- having in mind the wider scope of LASs, the proposed approach cannot succeed without stakeholder acceptance and support across society.

Recent research [2], as well as the conclusions from the FIG 3D Cadastres Questionnaire: Status in 2018 and Expectations for 2022 [16], show that countries are at different stages of 3D LAS implementation. Some of these countries have semi-operational 3D LASs, others have still no interest in introducing a 3D LAS, while there are some that do not have yet a (2D) operational LAS.

As illustrated in Table 1, numerous requirements and considerations need to be taken into account, to develop a well-operational 3D LAS, with regards to organisational, legal and technological aspects. The main objective is to achieve communication between the phases of the lifecycle, moving to automated processes and standardised models, as well as related methodologies.

**Table 1.** Requirements and considerations for the implementation of the vision of 3D LAS.

| Perspective Aspects | Requirements and Considerations | Description |
|---|---|---|
| **Organisational and Institutional** | Identification of users | There are various users involved that must be identified and their needs investigated (e.g., public, land registries, land surveyors, notaries, AEC industry, urban planners, local government, real estate agents, contractors, banks, valuators, engineers who issue permits, etc.) |
| | Political will and public demand | Governmental initiatives and eagerness to adopt a 3D LAS are crucial. |
| | Identification of relevant institutions | Involved institutions must be identified, including their level of involvement, and possible overlapping responsibilities. Engagement campaigns to educate and convince the stakeholders must be organized. |
| | Satisfactory level of interoperability | Interoperability and collaboration between organisations shall be enhanced using standards, while data exchange mechanisms must be established. |
| | Terminology, concepts and semantics used by different organizations to be clearly defined | Similar concepts may be termed differently and need to be organized within a semantically enriched structure (e.g., ontology), while new terms related to 3D aspect may need to be introduced and defined accordingly. |
| | Improvement of current workflow for registering an object | Estimation of the time and cost of current workflow that is expected to be improved when implementing the vision for LAS |
| **Legal** | LAS legal type | Type of LAS (titles, deeds, strata titles, other) to be analyzed. The level of maturity and current status to be investigated |
| | 3D parcel definition | The definition of "3D parcel" related to 'space' (including land, water, air & underground space) and not to 'land' is crucial and is an important step towards the implementation of the vision of 3D LAS |
| | 3D legislative framework | A 3D legislative framework is required, and there is a need to review and update existing regulations and laws to serve the needs of such a system |
| | Types of 3D objects | Identify the 3D objects' types to be registered and provide legal provision for these types. |
| | Legal mandate to comply with standards | Establishing as legal mandate to adopt or comply with standards at model level (national, European or international; such as: LADM, IFC, CityGML) |
| | 3D Public Law Restrictions | Introduce 3D Public Law Restrictions (PLRs) [59] when establishing or updating the 3D legislative framework |
| | Data quality | The desired data quality to be achieved in each phase of the system needs to be mandated |
| **Technological** | Compliance with standards | Currently custom-made methodologies and tools may be used, which are not based on (international) standards, Moreover, usually, there is no protocol for data exchange between different organizations and software packages and the exchange is based on files, which often results in data loss. Current databases, data elements, models and services are used to store and disseminate information: dependencies from software vendors and compatibility degree between data models |
| | Establishment of procedures | In most of the organizations usually, there is no clear procedure for data update and management of temporal objects (if any). Procedures, when exist, are manual and time-consuming |
| | Minimization of incompatibilities between systems and organizations | Similar datasets or different versions of datasets that currently exist in various organizations and contain incompatibilities (names, geometric representations, spatial dimensions (2D and 3D), and the attributes of the same physical objects vary between the different systems) |
| | Control of data quality according to the source data and the end product | Factors that affect data quality in terms of technological aspects:<br>■ different data acquisition techniques are used, which lead to different data quality of entry data,<br>■ datasets may be in local coordinate systems (i.e. BIMs) or they are not geo-referenced at all (floor plans),<br>■ different types of geometric primitives used, and validation rules need to be established,<br>■ Topology, or validity of objects (intersections and gaps) may not be maintained in the datasets and validation rules between each phase of the system shall be established. |
| | Qualification of personnel and determination of budget to be spend | Personnel must be qualified to use advanced technological tools & methodologies. Involved users must be able to adopt & use such resources affording training and meeting other costs |

A well-established and stable LAS is typically built around the organisational mandate which is driven by the public and stakeholders' needs. Such LASs are typically sophisticated in terms of their integrated organisational workflows and multi-faceted with respect to data entities and the relationship between them [60]. Most of the systems record interests associated with 2D and 3D parcels, buildings and condominiums, while the registration of the utility networks is being identified more and more often as a necessity.

The vision for a well-functioning, effective, 3D LAS in the future, is to be able to collect, store and visualise information for all those types of 3D objects and to record the 3D RRRs attached to them. It is evident that, due to their geometric complexity and thematic variation, the sources of the representations of those objects are multiple, and they are stored and exchanged in various formats. Therefore, it is necessary to consider the needs and requirements for modelling, storage, and visualisation of those 3D objects' types, when developing the proposed system architecture of a LAS in the context of a spatial development lifecycle.

## 4. A Vision for a Future 3D Web-Based LAS

Given the background presented in previous sections, it is envisaged that a future 3D LAS shall address, *inter alia* the above-mentioned requirements and considerations, in line with Spatial Data Infrastructures (SDI) best practices. According to INSPIRE [32], in the context of SDIs, data should be collected once and kept where it can be maintained most effectively, while it should be possible to combine seamless spatial information from different sources and share it with many users and applications. This can be best achieved with web-based information systems or in other words, Service Oriented Architecture (SOA).

This section is organized as follows: Section 4.1 introduces the key-features of the proposed System Architecture of a future 3D LAS, Section 4.2 describes a prototype 3D web-based LAS, while in the last section, it is illustrated how BIM/IFC data can be used efficiently as input in a 3D web-based 3D LAS, following the object lifecycle approach.

### 4.1. Key Features of the Proposed System Architecture of a 3D Web-Based LAS

In order to establish a system that manages spatial and non-spatial data (RRRs, land use planning, valuation information, etc.) in a consistent and coherent way, an appropriate system architecture is needed. Several components should be identified, including: 1. available datasets and datatypes, 2. a method of storing and structuring data, 3. acquisition and exchange of structured data, as well as 4. data visualisation and manipulation. Therefore, for the proposed system architecture, four components are determined, as illustrated in Figure 4.

A vision for a complete 3D LAS has been described in detail in the previous section, however, in this paper, emphasis is placed on the reuse of IFC data as input for LA purposes. It is expected, that the submission of BIM models in an IFC format would allow for data to be digitally archived, remain available and accessible in the long term and be stored in a machine-readable data model (in contrast with 3D PDF files).

Starting from the first component, the source data can be classified into three categories, considering their purpose. All source data, or at least a majority, shall be submitted via web-services. The data are classified as below:

1.  data collected using acquisition methods where various formats may exist depending on the method used (i.e. .las/.laz for point clouds, GNSS Receiver Independent Exchange Format (RINEX) data and/or .dxf and .shp for land surveying, orthorectified images and DTM (Digital Terrain Model) from aerial acquisition, etc.).
2.  data originated from design processes, referring to existing or future infrastructure elements. In this category, .shp and .dxf drawings for 2D data are the most common source formats, while IFC files are commonly used to store and exchange BIM models,

3.    spatial and non-spatial data from notaries, surveyors, land registries and cadastral authorities, may exist in various formats from paper-based and scanned pdf documents, to databases or ePlan files and digital maps (CAD-based or GIS).

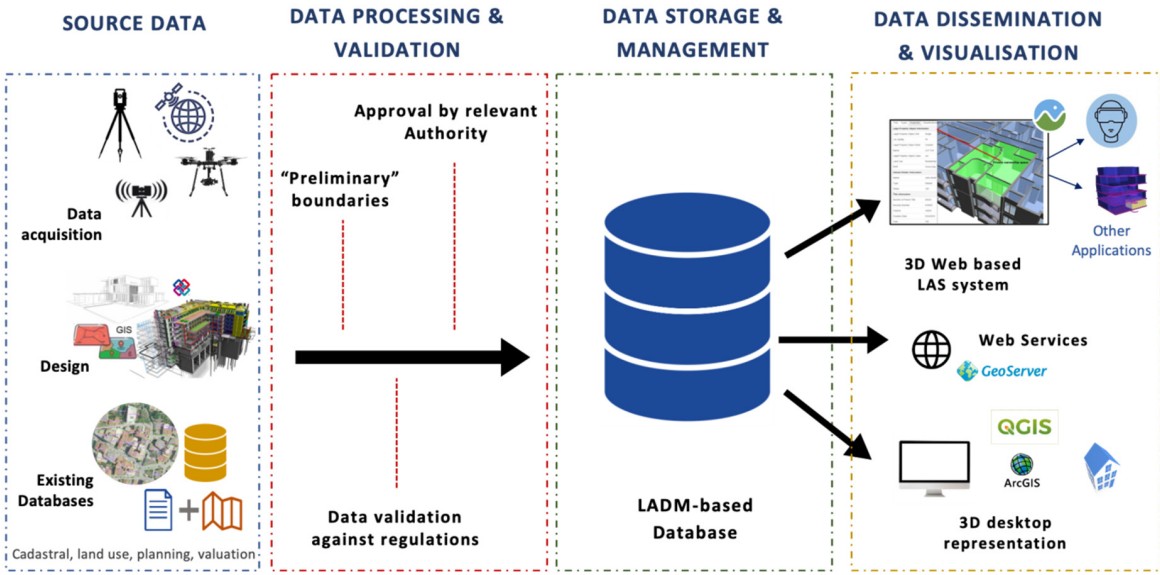

**Figure 4.** Proposed web-based system architecture for a future 3D LAS.

It is noted that for the available datasets and their geometric types, topological relationships, and attributes; issues regarding georeferencing, local coordinate systems, encodings, and terrain related issues (e.g., alignment of footprint with terrain, create terrain surface considering input point cloud or grid data, etc.) should be taken into consideration.

The second step is about data preparation, data processing and data validation. In order to establish the right connections between different data structures and their properties and to integrate and harmonise them, standardized formats and protocols for data input, storage, visualization, and dissemination should be used. It is expected that the geometry and semantic information from both GIS data and BIM models will be retrieved and stored in the database for such a system.

The combination and integration of the different types of source data, while keeping their semantics, geometry, and attributes is crucial. This preliminary phase will result to the "initial boundaries"; i.e. an initial version of the object's legal boundaries derived from the physical ones. At this stage, the input data will be examined in terms of data quality and also the Level of Detail, and if applicable (depending on the required LoD) they will be simplified. What is more, as physical boundaries do not always coincide, or do not have to coincide with the legal boundaries, a validation step is essential. Therefore, the next phase is data validation against the necessary regulations and building codes, as well as verification and approval from juridical experts of the legal boundaries. This is an iterative process, where the input data will accordingly be processed in order to meet local/national regulations and to be approved by the relevant national mapping authority/notary. It is expected that validation can be integrated in workflows fully automatically in the future by using artificial intelligence.

Structuring of identified spatial and non-spatial data is a key component of the system architecture; thus, the third component is data storage. Explicit geometries and/or topological relations from the source data will be retrieved and thus, different data structures containing different properties and characteristics should be stored together. This requires that the database will be based on a generic (national) database schema and comply with LADM concepts, terminology and structure.

A spatial database management system (such as Oracle Spatial, PostgreSQL/PostGIS etc.) should be selected to provide comprehensive support of various extensions supporting import, spatial indexing, manipulation and restructuring different spatial data types such as points, lines, polygons,

solids and point clouds. Mechanisms to validate the compliance of database schema with LADM concepts, vocabulary and structure need to be further developed.

Once data is organized in the LADM database, there are various visualization approaches that can be followed depending on the end product and the user needs and requirements. Data visualisation is an important component of the system architecture, providing an option to users to interact with the data. To establish the connection between the database and the visualization platform there are various directions to be followed and explored (e.g., Geoserver for streaming simple geometries, custom-made methodologies using international standards for more complex geometries, dedicated libraries and formats for extruded buildings, BIM models and other data into 3Dtiles, such as py3dtiles [61], etc.). The following visualisation and dissemination approaches may be followed (keeping in mind that Registering in a LAS is one phase in the object lifecycle, with subsequent phases following, such as Maintaining and Valuating, which need input data from previous phases):

1. development of a 3D-based visualisation platform to disseminate and query the data (Cesium JS platform, etc.). Such a platform should support the visualisation of different spatial data types and may also include various tools, such as: splitting apartment rights online, that can be managed by various users (e.g., notaries), while further applications may be developed in the context of the spatial development lifecycle approach (e.g., Virtual Reality application for underground utilities);

2. provision of 3D web services to disseminate the data in various formats within the specifications of the National Spatial Data Infrastructure or the National Geographic Information Infrastructure of each country;

3. export data to be visualised in a 3D desktop environment (QGIS, ArcGIS, FZK viewer, etc.).

Summarizing, the described process is an iterative one, as data dissemination can be used as "feedback" to support the data acquisition, design etc. Therefore, the output is used in support of the input. For instance, imagery can be used as background to GPS based boundary data collection; existing boundaries can be used to support collection of new boundaries, etc.

*4.2. Prototype of a 3D Web-Based LAS*

This section presents the on-going development of a prototype 3D web-based LAS [62], currently with test data from the Queensland Digital Cadastral Database (DCDB) [46], which is developed on the basis of the vision for 3D LAS as presented in the previous section. The web-based approach is used to support the full object lifecycle with phases before and phases after the actual Land Administration. The prototype combines different types of data to give context to the 3D cadastral parcels, namely:

■ 3D survey plans, both 'building format units' and 'volumetric parcels';

■ 2D cadastral parcels;

■ Registration of rights, restrictions and responsibilities (RRRs) and parties (with falsified names and details for privacy reasons);

■ Elevation data (DTM or DEM, depending on data availability) in order to make the visualization more complete and meaningful (courtesy of Fugro);

■ Reference data such as topographic objects, either in 2D or 3D.

The tools used include the PostgreSQL/PostGIS database, the Apache Tomcat web server, the GeoServer WFS server, the Cesium JS 3D web viewer (WebGL based), and a range of custom software; e.g., to convert the 3D survey plans semi-automatically to 3D parcels for the database (as presented in Section 4.3. For research purposes, a significant subset of the current DCDB (which is stored in an Ingres database), has been loaded into PostgreSQL/PostGIS (more than 3 million parcels, including history).

For the prototype database, the basic tables of the current DCDB (in Ingres), have been converted to PostgreSQL/PostGIS, with no change in logical structure. The main additions for the prototype

are a table of faces, used to represent the boundaries of the 3D spatial units (parcels) and table to store survey plans, parties and rights. In order to give a good selection of 3D spatial units, a number of parcels in the suburb of Kangaroo Point have been manually or semi-automatically derived from survey plans. It was decided to keep the database in a form equivalent to that used in the current Queensland DCDB, but to expose views of that data in a form which is compatible with the LADM. This achieves four purposes: 1. It allows simpler loading of future data from the Queensland DCDB, 2. It allows modification of the prototype database structure without invalidating work being done on the visualization, 3. It provides the possibility of defining an LADM derived protocol for delivery of mixed 2D/3D+t Cadastral data, and 4. It indicates that a database which is not defined with LADM in mind can still support such a protocol. Figure 5 shows the LADM compliant database schema of the 3D LAS prototype with original Queensland DCDB tables in light pink, new tables in orange, and 'views' in blue.

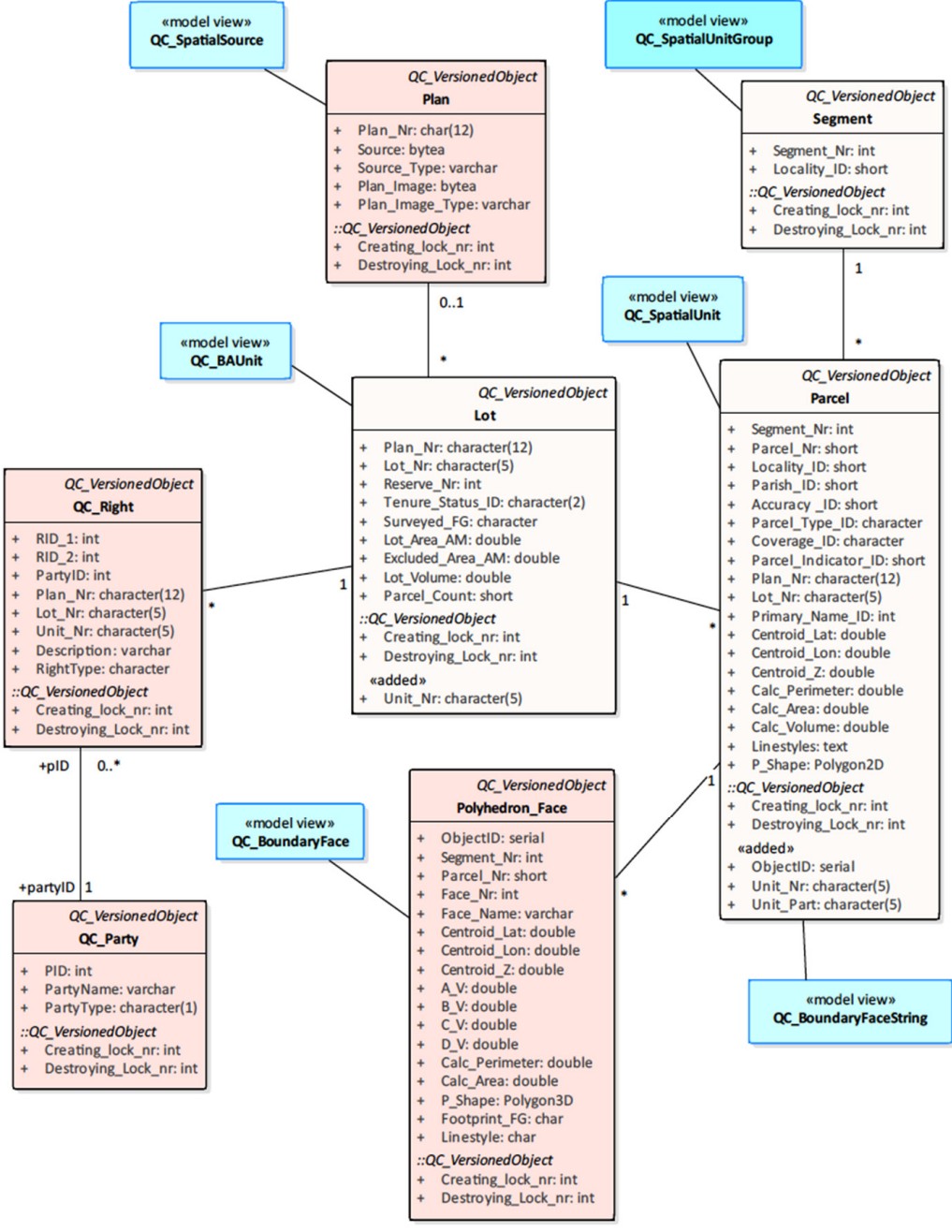

**Figure 5.** Database schema of the 3D LAS prototype [46].

### 4.3. BIM/IFC Data as Input in a 3D Web-Based 3D LAS

In this section, recent work on how to use BIM data as input for LAS, focusing on apartment rights, from Meulmeester [22] is presented. The intention is to realize a more efficient flow according to the object lifecycle thinking from design to registration phases, compared to the costly semi-automatic conversion of survey plans. It should be noted that there is also previous research exploiting the usage of BIM/IFC data as input for LASs, namely: Oldfield et al. [63], Oldfield et al. [18], Atazadeh [20] and Atazadeh et al. [21], which have been taken into account.

Meulmeester [22] proposes a proof of concept of a complete data processing chain for registering new apartment rights in 3D in the Netherlands, by enriching IFC files with legal information.

At the proof of concept developed, the steps that have been followed are (Figure 6):

■   BIM legal creation: the IFC model is enriched with legal information by designing a user defined property set with cadastral information, added to the 'IfcSpace' element. The 'Cadastral Information user defined property' set contains the required information to register the spatial representation of apartment rights in 3D. The current (Dutch) regulations w.r.t. the requirements for the 2D apartment floorplan drawings are projected on a 3D representation, which results in the contents of the cadastral information user defined property set.

■   Automatic extraction of 3D legal space for registration of apartment rights, by performing mapping between IFC entities and LADM classes.

■   Validation in terms of correctness and completeness. A set of rules for IFC files enriched with legal spaces has been developed, while checks were also performed in the database (overlapping geometries, completed user defined property sets, etc.)

■   Storage in an LADM compliant database (both IFC geometry and attribute data).

■   3D web visualization for dissemination purposes on the Cesium JS platform and desktop visualization using QGIS

■   (sub-) Splitting and merging existing 3D parcels in a building have been introduced as functionalities to change cadastral information on apartment rights (Figure 7).

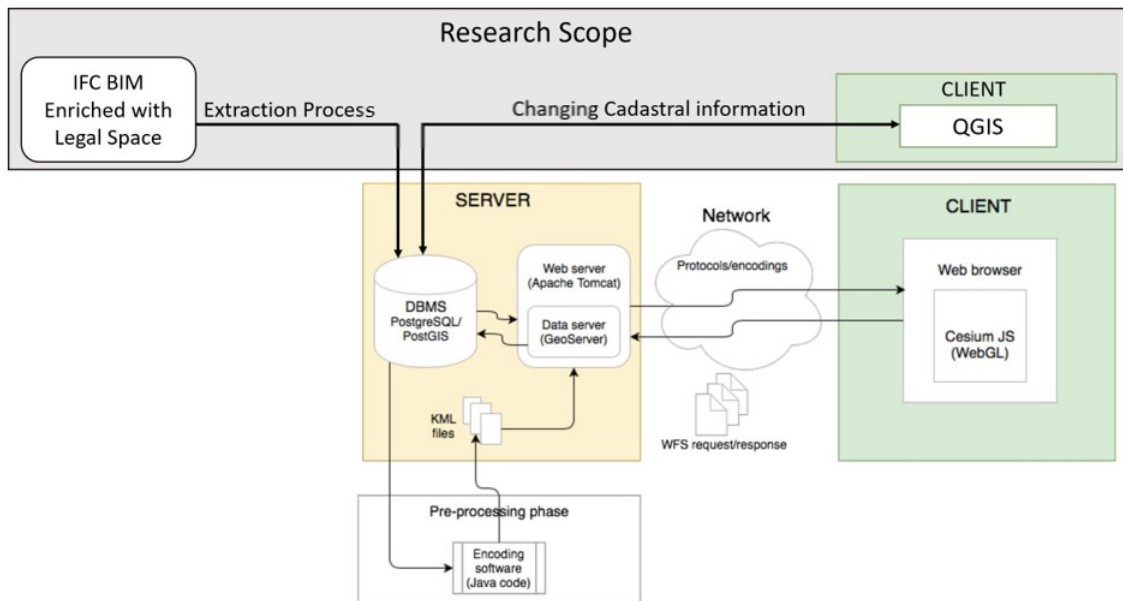

**Figure 6.** Architecture of the BIM-Legal prototype [22].

A similar approach, with regards to enriching IFC files by adding Property Sets within the IFC file, has been recently proposed by Olfat, et al. [19], where a BIM-driven building subdivision workflow is presented.

The proposed web-based system architecture for a future 3D LAS, is based on the principles of data reuse and interoperability, thereby attempting to validate the proposed methodology and introducing a whole of life approach. Moreover, this solution fits within the current standards of the buildingSMART consortium and can be communicated to users with an Information Delivery Manual (IDM). A proposal for the Dutch IDM has been developed to show how the proposed solution of defining legal space can be considered and put in practice by BIM creators. This is a realization of the lifecycle thinking: information from one phase efficiently flowing into the next.

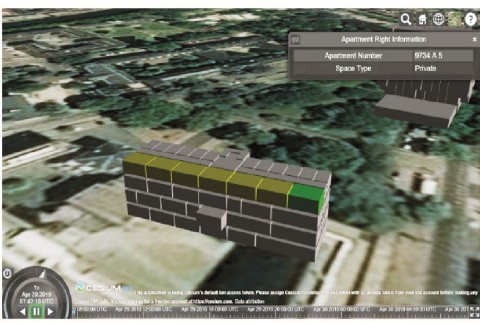

(a) Office building with apartment '5' before the split

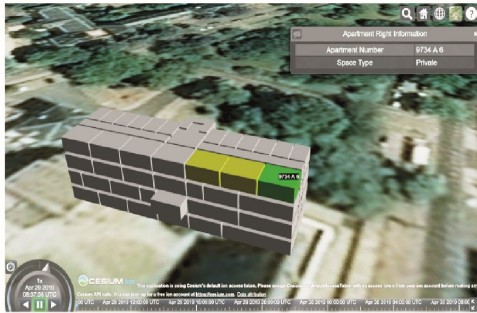

(b) Apartment '6' highlighted

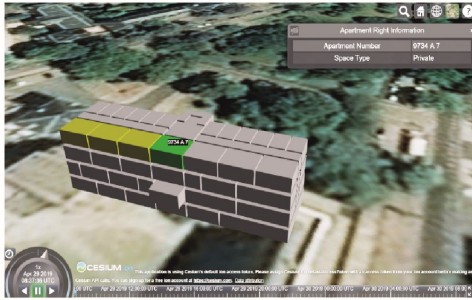

(c) Part of apartment '7' highlighted

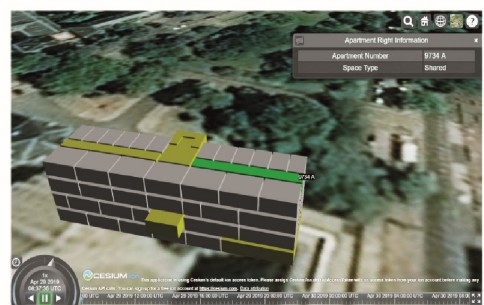

(d) Part of shared space of building complex '9734 A' highlighted

**Figure 7.** Visualization at the 3D web-based prototype. Example of office building splitting apartment 5 (**a**) into apartments 6 (**b**) and 7 (**c**) visualised in Cesium JS. The shared space is highlighted in (**d**) [22].

## 5. Conclusions

The paper considers the future of 3D LASs in a wider context and as an important phase in the Spatial Development LifeCycle, with regards to legal, technical and organisational aspects. Recognising the implications that result from the constantly changing environment and the emerging technological advances, the needs and considerations of interactions between SDC and 3D LAS are addressed in the paper, with a view to discussing a cross-sectoral approach to collect, maintain, re-use and share 3D data.

A 3D LAS is a core component of the approach presented. Its potential role and key features in the context of the full lifecycle is discussed. There is an emerging need to move from 2D to 3D LAS in this context. A brief reflection on the so-far developed LASs around the world and the 3D objects that are being registered is presented. Aspects that need to be further addressed to develop a well-functioning 3D LAS in a wider context are also stated.

A cross-sectoral approach to the process of collection, maintenance, re-use and sharing of 3D data can improve the efficiency of today's data management processes. The data may be usable for various applications. The semantics and the exchange format of 3D information is demanding greater engagement with external stakeholders (outside Land Administration process/activities). At the same

time new technology advances and standards allow meaningful collaboration. Land administration can go beyond the traditional legal mandate of the organisations as land registries, and custodians of cadastre. Land administration can be applied to meet broader interests in the spatial development lifecycle with well-structured workflows and with conformity to standardisation. Requirements and emerging challenges for the development of such a workflow are presented in Section 3.3. Those requirements and challenges are the foundation for the proposed system architecture of a future web-based 3D LAS which is in line with SDI good practice. This system architecture is presented in Section 4. It establishes a system that manages spatial and non-spatial data (on land tenure, land use planning, land valuation, etc.) in a consistent and coherent way. The ISO 19152:2012 Land Administration Domain Model (LADM) in its current Edition I, as well as in its (under-development) Edition II is proposed to serve as the core structure of the 3D LAS, while the central role that BIM plays in the life cycle of development projects is recognized. Emphasis is given on making feasible the reuse BIM/IFC data in a 3D LAS.

The proposed system architecture has four main components, including 1. source data gathering, 2. data processing and validation, 3. data storage and 4. data dissemination and visualization. Technical, institutional and legal aspects are considered. The suggested architecture supports streaming of a large number of different datasets, using international standards, into a unified/joint data structure. As the use of BIM is becoming mandatory by more and more governments it is considered as a promising data source for LAS. The proposed system architecture as described in Section 4.1 is intended to be a 'blueprint' for replication. Therefore, Sections 4.2 and 4.3, describing the development the prototype of such a future 3D Web-Based LAS, contain quite a lot of details about the data model, data sets, and tools. The source code of the BIM legal demonstration is open and can be found at https://github.com/TUdent/BimLegalDemo. However, a holistic solution for a well-functioning 3D LAS requires simultaneous research and advances in both the legal and technical aspects [12,49]. In the presented research approach, there are several technical challenges to be resolved. Those are part of future work of this research. For example, validation of input data in the various stages of the system architecture implementation and the desired level of accuracy for each registered object are amongst the important issues to be further explored.

## 6. Future Work

3D LASs will be well established and used at global context in different levels of maturity – depending on the jurisdiction and political will. Current practices evolve in line with advances in land development, infrastructure expansion and smart cities. In this regard, the ISO 19152:2022 LADM Edition II, serves as the enabler. The second edition of the standard adopts a holistic approach. Its proposed extended scope includes spatial planning, valuation information, etc. It provides a standardised approach for a complete 3D LAS as a component in the spatial development lifecycle. As the revision of the standard is ongoing, the future developments of LADM edition II, as a multi part standard should be considered at the proposed system architecture's elements.

What is more, it is expected that, eventually, 3D property information will often be created in the design and construction phase, thus data reuse in the lifecycle of an object will become the new trend. More BIM/IFC models representing various types of 3D objects will be available in the future (e.g., IFC for linear infrastructure: IFC Rail, IFC Tunnel, etc.). The proposed prototype shall be tested to examine its performance with those objects and will be updated accordingly. For other objects to be registered in such a system, the functionality and performance of other standards should be tested. For example, LandXML and InfraGML, as well as the new version of CityGML (3.0) supporting compatibility with LADM.

From a technical perspective, future work includes further refinement and more explicit and detailed study for each one of the components of the system with real-word data to validating the key features of the system architecture. A further step of this research is to investigate how to extract geometry and semantic information from IFC models and how to process and store them accordingly

at an LADM-compliant database. Visualisation in a web-based platform and specifically the 3D LAS web-based prototype using Ceisum JS (Cemelini et al. 2018) could be a next step.

In order to achieve the development and adoption of such an LAS, as part of the lifecycle (plan, design, approve, finance, survey, construct, register, use, maintain, demolish), international collaboration is a requirement. Interoperability is needed at different levels: at data level and at model level. Therefore, standardisation organisations, as well as initiatives aiming to establish and promote global policy frameworks for the production, availability and use of geographic information (e.g., UN-GGIM, etc.) need to be actively involved. Last but not least, the technology comes a long way and is able to satisfy the needs of such a system, while there is less research and proposed sustainable solutions, in terms of the institutional and legal issues.

Talking about lifecycle, information workflow, and reuse of information over time, the fourth dimension is relevant. There is a need to include 4D, meaning 3D + time, to reconstruct history, to manage events in maintenance processes, to reflect reality in case of temporal rights., to include spatial units with different accuracies, dimensions and representations over time, to access both current and historic versions, etc. The time dimension is also a crucial aspect in BIM modelling. Therefore, a future step should be to implement the proposed approach and the system architecture in 4D LAS, supporting efficiently the time dimension.

Beyond the scope of the current paper, but also fitting in the spatial lifecycle approach, is the support of the circular economy by registering the materials used in various constructions. The proposed 3D web-based LAS is a possible example for the set-up, organisation and architecture of these future Materials Cadastres, sometimes called "Madasters".

**Author Contributions:** This research is a result of the collaboration and contribution of all authors. Efi Dimopoulou, Christiaan Lemmen and Peter van Oosterom are responsible for the preparation, description and analysis of the terminology and the new concepts presented in Section 1. Eftychia Kalogianni summarized the state of the art in Land Administration (Sections 2, 3.1 and 3.2), while presenting the considerations for the implementation of the vision of 3D LAS in Sections 3.3 and 4.1, under the guidance of Efi Dimopoulou, Christiaan Lemmen and Peter van Oosterom. What is more, Peter van Oosterom contributed to the introduction of a 3D WebGIS prototype using BIM as input. All the authors have contributed to the Conclusions and Future Work, to provide food for thought for the next steps. The coordinator of the authors is Peter van Oosterom, and each author made a substantial contribution in the preparation of the manuscript. All authors have read and agreed to the published version of the manuscript.

**Acknowledgments:** The work that is presented in this paper is part of a wider research, an ongoing PhD study carried out at Delft University of Technology (TU Delft) covering the methodological framework and many of the key concepts. The authors would like to thank Rod Thompson for his careful proofreading and correcting the final version of the paper.

**Conflicts of Interest:** The authors declare no conflict of interest.

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
