# Peer review of "3D Land Administration: A Review and a Future Vision in the Context of the Spatial Development Lifecycle"

_ijgi, doi:10.3390/ijgi9020107_

Round 1

Reviewer 1 Report

This is an interesting topic. Even 2D LA is used as the core for SDIs.

My comments to improve this paper:

I suggest to review the Abstract. It is not easy to follow, specially when the topic of discussion jumps from collaboration and lifecycle-thinking (Para 1) to 2D and 3D LA (Para 2). No connection is there. Your Abstract provides different background, however does not articulate the current challenge/problem that you want to solve?

If I ask you what problem you found that motivated you to write this paper, what your answer is. This needs to be reflected in the Abstract.

BIM aims to play a central role in the life cycle of developments. It is not well described in your work what the additional value 3D LA adds to this. Is it just a component of BIM? or ...

The aim of this paper is not clear to me.

I expect more a precise and scientific Abstract to understand your question, aim, and approach to achieve your aim to be able to understand the rest of your work. 

Author Response

see pdf

Reviewer 2 Report

General comments:

I found the paper difficult to read, the message is not clear, what is exactly addressed by this work (moving from 2D LAS to 3D LAS? The integration of domain instead of data (so break the silo approach)?), state it more clearly in the section introduction. The authors talk about many concepts, they are not well defined and it is difficult to understand what is the value of the discussion and thus the paper. But I do believe that there is interesting material in your thinking, but it has to be explained and better presented. Hope my comments will help the authors to improve the paper.

In fact, I am interested to get your point of view on LAS, and its role in relation with new society trends such as 3D geospatial systems and data, Internet, Big Data, AI, and even societal priorities as climate change, sustainable development, environmental issues, etc.  You can even try to address the simple question: Is LAS will still exist in 2030 ? To address this topic and try to find answers, literature review is a first step, but you should use other methods as having interview with experts and/or authorities, having an online questionnaire, etc.

For instance, methodological aspects have to be explained (how this discussion is enabled by literature review only? by group discussion? How it is validated by cross comparison? why this web-based system architecture? How it is validated? What are the scientific questions addressed, what is the hypothesis supported by the authors, etc). When we search the web, it is quite easy to find that this work is part of a Ph.D. study, so methodological aspects have yet much more importance. For example, it could be mentioned in the paper that this work is part of research project which aim to ...

What are exactly the objectives of this work? It is not clearly presented. Line 43, it is indicated: “This paper emphasizes the potential role and key features of LAS in the context of the full lifecycle of spatial objects and proposes a web-based system architecture for a future LAS, based on the principles of data reuse and interoperability.” The first part is ok to explain a bit the objectives, but developing a Web-based system is not an objective, it is a method to validate something at a higher level, but what?

Better express the outcomes (expected results).

Is LAS=Cadastre ? for me not, and I do believe that most of the paper is talking about cadastre not LAS.

At the beginning of the paper better define the concept of LAS (with relevant references like those published by Stig Enemark or Ian Williamson, so not only cadastre references).

The concept of Spatial Development Lifecycle is not well explained and has to be, since it is the core concept in this paper. And more, I am not sure if this expression is correct, spatial development? What is it exactly?

I am confuse: Are Object Development Lifecycle and Spatial Development Lifecycle the same concepts? The authors seem to use both terms without distinction, for me, there is a distinction. What is Object Development? Object is referring to what ? not clear.

I do not understand why you make the link between LAS and BIM and not LAS and CityGML (that is an OGC Standard). The authors seem to have taken some decision to orient this work, but those decisions are not explained.

Conclusion should summarize the outcomes, the results, the contributions of you work, to be revised.

Detail comments:

The abstract is more written as an introduction, we should find more easily the outcomes, the expected results of this paper. Only the last sentence, lines 43-45, is providing a bit of material but we need more (based on line 43-45) what are the results of your work? Explain it, in the abstract.

Be aware, I found a large number of statements not really supported by references or concrete examples as the third paragraph in the section Introduction, first sentence of the fourth paragraph, sentence starting at line 85 (the impact of ...), sentence starting at line 133, etc.

For instance, in the section introduction, it is missing reference and explanation about why LAS and cadastre system should move to 2D to 3D. Yes, I found some of these explanation is section 3.3 but it arrive too late, we need to get those information in the section introduction (since moving from 2D to 3D is central to this work, isn’t it ?).

The reference van Oosterom 2018 is a book with various sections, so when referring to this book, I suggest to be more precise (which book sections, where can we find discussion and who is talking about the related topic).

In the sentence starting at line 59 (The majority of existing LASs and cadastres...) is missing the term data or system (2D what?)

A number of sentences are long (as the first sentence of the fourth paragraph) and it does not help to understand the message.

Line 91, IFC is interesting yes, but not because it is a format, because it is a formalism (a language), semantically rich.

Lines 43 and 102, add the notion of 3D in the sentences since this concept is a key driver in the proposed paper.

The formalism used for Figure 1 is not clear (there is a big rectangle that seems to indicate that land administration, which is connected only to registering, I do not really agree, see the reference of Enemark or Williamson)

I do not exactly understand the link between the discussion of LAS in the section Introduction and the discussion about Spatial Development Lifecycle in the next section. Better link up one paragraph to another.

There is a format error at line 140 (in fact, all the references to figures are not correctly formatted).

Line 184, data sharing is more related (and challenging) to the fact that data is collected for one purpose and finally used for another purpose (e.g. I collect the location of this road because I want to produce a road map, and next someone else will use this road location to estimate city zoning regulation). It is not a question of using the data many times as mentioned by the authors. Give clear example, it will help.

Section 2.3, please explain the difference between standards and protocols.

For me, BIM (Building Information Modeling) is an approach and BIM (Building Information Model) is the output product (3D AEC models), but not really an official standard as stated by the authors line 235. IFC is a standard. For example, ISO is providing standards for managing information with BIM. BIM will benefit from standards, but BIM itself is not a standard.

Line 248, if the authors say BIM is not new, so then explain when it was first mentioned, by who, in which context, etc.

Figure 2, I found the same picture here http://futureofconstruction.org/content/uploads/2016/09/BCG-Digital-in-Engineering-and-Construction-Mar-2016.pdf, a report by Boston Consulting Group 2016, and in other reports, so the source of the figure has to be double checked. And when you say that it is adapted, it has to be explained (for now, it looks exactly the same).

Section 3.1 be clearer of what bring in the third dimension in LAS. Aien reference is a good one but as you know there is many others. Many authors state that the third dimension of legal situation is already managed by cadastre system, it is simply not represented in 3D (with spatial data). Be clear about the specific challenge that pose having the third dimension for LAS (not only cadastre).

3D Cadastre = 3D LAS ?? not sure!

Section 3.1. you are certainly aware of the UN- Expert Group on Land Administration and Management, maybe refer to this reference, it presents 9 pathways for effective LAS, and could be relevant to discuss it in your paper (available here: http://ggim.un.org/meetings/GGIM-committee/9th-Session/documents/E_C.20_2020_10_Add_1_LAM_background.pdf)

Figure 4, which originate from Williamson, use a triangle to express and oppose three aspects (based on this triangle shape, the best LAS should be then evenly distributed between all three sides). From my point of view, the triangle is not the correct shape to express the relation between these three aspects. Today, the technical aspects is not a real challenge, and we do not have to oppose technical aspects with legal aspects, neither with institutional aspects. Anyhow, authors should explain why they decided to use this figure and discuss how this figure elucidate their understanding/definition of what is a LAS ?

I much prefer the figure of Enemark 2004, which express the paradigm of Land Management and it includes concepts as land policy, economic, social and environmental.

Section 3.3 only contains objects about cadastre not about the larger concept of LAS (for example, urban georegulation have to be part of it, infrastructure as road are also object to be considered).

Table 1 is an interesting outcome, should be more developed.

Section 4, I do not really understand why the authors are presenting such development, it has to be explained. You try to demonstrate what?

Author Response

see pdf

Round 2

Reviewer 1 Report

I see changes in the Abstract section, unfortunately it has not improved the content.

Honestly it is not easy to follow the content of this work. Line 31 starts with "Given this background"; 30 lines provided as background to address the research question in Abstract section. This shows that the research question is not clear or cannot be clearly articulated. you can use this background in the Introduction section.

I did read the Abstract several times; and guess that the research question is the importance of LA systems in the development lifecycle, so the question can be "what the role of a LA system is in the development lifecycle?". Why is not this simply mentioned? why are there many lines for background? My question is what happens if I remove line 13 to 30? To me nothing it would not impact the content yet it would simplify the abstract.

My suggestion is to review the paper and make it simple for the readers. 

Line 19: I could not understand what this means:"Currently, the disciplines involved in those phases are quite autonomous, yet, they are constantly moving to adopt 3D data techniques and use 3D technologies."

The aim of a paper targets normally to solve/answer the research problem/question. First of all the aim has not been simply written. Secondly how your mentioned aim solves your problem?

I appreciate the work of authors in this paper, unfortunately the content has not been well and scientifically presented.

Author Response

see attached file, section 'Reviewer 1'

Reviewer 2 Report

I like the revised abstract. I found it a bit long, but will depend on the journal guidelines.

I like the section Setting the scene (in some way, it is part of the contribution of this paper).

All the definition of Land administration should be placed in this section and not later (we found alternate definition in section 3), there is a bit of redundancy. And is the authors make a difference between land administration compare with land management?  Enemark always talked about land administration but the figure 4 which refer to the work of Enemark is marked as land management (not land administration).

Line 154, what is this “wider approach” could you explain it a bit? (in order to understand where the content of this paper fits in this wider approach).

Line 157, remove “as already mentioned”, since the following information is new (abstract does not count).

Line 171-172, the Web based system will be used to validate the approach but to validate which aspects (ex. the applicability, the value added for the user, the completeness of concepts, the rigorousness, etc?)? It is not explained, neither in section 4.

Line 158 and in other places in the text, the authors mentioned that they will propose an approach, but where is this approach in the paper? Line 190 it is written that section 2.3 may propose the approach but section 2.3 is entitle The importance of Standard. Is it the Web-based system architecture section 4 ? Not clear. At least, one section should be called Proposed approach, in order for the reader to easily find this approach. So, be clearer about what and where is the proposed approach.

Line 548, I think it is missing references after Singapore (REF)

Lines 572 to 583, it is a discussion more relevant for section 3.4 that is about requirement of 3D LAS.

Table 1 could be an interesting contribution of this paper, but currently, I found it a bit confusing. For example, you place Users as requirements/challenges, I do not understand. If the message is to provide a clear view of what we have now in term of LAS and what we need to have a well operational LAS, placing requirements and challenges in the same column is not the best way to express it. Are you able to distinguish in separate columns ?

Figure 5, since the authors talked a lot about BIM data and thus to be clearer for reader, I suggest to add (e.g. BIM) in brackets after Design. For this architecture, OLAP systems could even be take in consideration since it quite similar to your architecture (i.e. having an intermediated, aggregated levels of data, validated to answer specific queries).

Line 666, having "relational" database in not required, but any spatial database management system.

The web based system proposed is impressive. I am curious to know more about the replicability of the work done by the authors, should be discussed in the conclusion.

Before having the discussion about future work, we need a conclusion. Express the outcomes of this work, the contributions. I do believe that your work propose interesting outcomes but the authors need to discuss them, it is missing this part, and it is important.

In overall, and since the text and discussion are long, we are now loosing a bit the real contributions of this work. I think that a final step to reduce and optimise the manuscript is required.

Author Response

see attached file, section 'Reviewer 2'
